# OlymBio-Bench: A Multimodal Challenge Towards Expert-Level Biological Reasoning

## Abstract

The evaluation of large language and multimodal models requires benchmarks that go beyond simple knowledge retrieval to assess complex reasoning, especially in scientific domains such as biology. Existing biology benchmarks fall short, either being text-based, too low-level, or lacking the integrative reasoning needed for expert-level problems. To address this problem, we introduce **OlymBio-Bench**, a novel, expert-level multimodal benchmark for biology. Sourced from over 220 frontier research papers in life sciences and curated by a dedicated team of over 60 authors and reviewers, our benchmark is uniquely challenging, requiring sophisticated inference and multi-step reasoning within realistic research contexts. A key feature is its inherent multimodality, with a large majority of questions incorporating essential images, diagrams, and data plots that demand integrated visual and domain-specific understanding. We evaluated a range of state-of-the-art models on OlymBio-Bench, and our results reveal that even the most powerful models fail to achieve a passing score, highlighting critical deficiencies in their ability to perform complex, multimodal scientific reasoning. We further demonstrate a strong correlation between question complexity and model failure rates, with multimodal questions posing a more significant challenge than text-only ones. Our findings confirm that OlymBio-Bench is a formidable and unsolved challenge that can serve as a critical resource to catalyze the development of next-generation AI models capable of more advanced biological scientific reasoning.

## 1 Introduction

The rapid progress of large language models(LLMs) creates an urgent need for rigorous evaluation. Benchmarks are not only for scorekeeping; they diagnose weaknesses, track meaningful progress, and guide the design of models. Besides, specialized benchmarks are crucial to the development of subject-targeted models. Biology is a particularly demanding domain: beyond factual recall, it requires integrating disparate information sources, interpreting complex experimental data, and executing multi-step reasoning. As models are increasingly tasked with higher-level problem solving, there is a clear need for benchmarks that mirror authentic research complexity and assess expert-level scientific reasoning rather than simple knowledge retrieval.

Existing life-science benchmarks have begun to evaluate the reasoning of models and their ability to handle realistic biological scenarios. Recently, researchers have also emphasized visual question answering and chart/figure understanding, but these efforts typically target general-purpose perception or under expert-level content rather than domain-specific expert-level biology. In contrast, OlymBio-Bench is purpose-built to fill this gap by combining (i) research-style, expert-level biological problems, and (ii) visual inputs that are essential to solving the tasks.

To meet this need, we introduce **OlymBio-Bench**, an expert-level, multimodal benchmark for biology. OlymBio-Bench is built from an original self-designed collection of problems themed on the International Biology Olympiad (IBO) that situate complex questions in realistic research contexts. The problems span many subdisciplines and require sophisticated inference. Crucially, the benchmark is inherently multimodal: a majority of questions include essential figures (images, diagrams, plots) that require visual understanding tightly integrated with domain knowledge.

The evaluation results reveal a clear performance hierarchy in which the newer models oriented to reasoning outperform the predecessors, indicating that the benchmark effectively captures meaningful

capability differences. At the same time, even the strongest models fall well short of perfection, underscoring persistent gaps in multimodal biological reasoning. We release OlymBio-Bench as a resource to catalyze the development of next-generation models that reason more like human experts and can better accelerate biological discovery.

## 2 RELATED WORK

### 2.1 THE DEVELOPMENT OF LARGE LANGUAGE MODELS

Over the past few years, LLMs have advanced rapidly—from "follow directions" to "think and reason" to "search deeply and synthesize evidence". The first milestone was the rise of instruction-tuned models, which use supervised fine-tuning and human feedback to reliably follow natural-language prompts. ChatGPT (OpenAI, 2022) popularized this paradigm, combining instruction tuning with context handling to deliver a better conversational experience. Soon after, models like GPT-4(OpenAI, 2023) expanded beyond pure text to handle multimodal inputs (e.g., images) and showed stronger logical consistency on complex tasks. Recent GPT-4.1(OpenAI, 2025a) also emphasize longer-context understanding and instrution following, improving performance on document-level tasks. There are also models such as Qwen 2.5 lines(Bai et al., 2025).

The next shift has been toward reasoning models, which allocate extra test-time compute to deliberate through multi-step problems before answering. Examples include GPT-4o(OpenAI, 2024), which are designed to use chain of thoughts methods, enabling more reliable solutions in math, coding, and analysis. Grok4(xAI, 2025) from xAI is also a popular reasoning model.

The latest trend is deep-research or agentic models—systems that iteratively search, read, and reason to produce evidence-backed syntheses or reports. Instead of "search once, then answer", these models loop through "search → reason → search → reason", cite sources, and converge on defensible conclusions. Representative examples include OpenAI's Deep Research approach GPT-5(OpenAI, 2025b) and Google's Gemini-2.5-Pro(Comanici et al., 2025), which pair web exploration with structured reasoning to deliver analyst-style outputs.

### 2.2 LLM BENCHMARKS TARGETED AT BIOLOGY

Benchmarks not only evaluate models; they also steer progress by revealing where capabilities lag. To rigorously assess whether contemporary LLMs can solve biological problems, biology-focused benchmarks are essential. Below we situate domain-specific efforts within the broader landscape of science benchmarks.

General science benchmarks such as GPQA (Rein et al., 2024) and SuperGPQA (Du et al., 2025) probe expert-level knowledge with carefully curated questions, but they are text-only and thus cannot test reasoning over figures, diagrams, or data visualizations that are central to biological inquiry. ScienceQA (Lu et al., 2022) aligns with K–12 curricula, limiting its utility for advanced assessment. MMLU (Hendrycks et al., 2020) includes biology but is predominantly undergraduate in scope, constrained to multiple-choice text, and misses the integrative, cross-modal reasoning typical of research settings. PubMedQA (Jin et al., 2019) targets biomedical question answering grounded in PubMed, requiring interpretation of quantitative results and statistics, yet it, too, remains largely text-centric. These scientific benchmarks surface important abilities but leave a notable gap for expert-level multimodal biological reasoning.

Among biology-specific resources, LAB-Bench(Laurent et al., 2024) is a comprehensive suite practical research tasks (e.g., literature search, protocol planning, data analysis). It evaluates abilities such as literature recall and reasoning (LitQA2, SuppQA), figure and table interpretation (FigQA, TableQA), and protocol design (ProtocolQA). Although multimodal, most tasks are relatively short and direct, offering limited stress on extended chains of reasoning. Other benchmarks tend to focus on subdisciplines—such as the multimodal virology benchmark VCT(Götting et al., 2025) and the computational biology data-analysis suite Bix-Bench(Mitchener et al., 2025).

In summary, current resources provide valuable snapshots of factual recall and text-only problem solving but fall short on comprehensive, expert-level biology questions that demand synthesis across heterogeneous evidence (e.g., experimental plots, microscopy images, statistical charts). This gap

motivates a benchmark that explicitly targets multimodal, expert-level reasoning. Accordingly, we introduce a benchmark designed to probe the limits of domain knowledge and analysis, enabling precise evaluation of factual competence, reasoning depth, and robustness on expert-level biology tasks.

# 3 METHODS

## 3.1 DATA COLLECTION

All questions were sourced from practice sets originally developed for participants in International Biology Olympiad (IBO, the highest-level event for biology Olympiad). These questions are covered by confidentiality agreements and access controls, and are not publicly available (e.g., via web search). In addition, every question underwent a structured curation pipeline—including drafting, expert review, and selection—the full workflow of which is shown in Figure 1.

Figure 1: Workflow of data collection

During item creation phase, three distinct roles were involved: the item author, the reviewers, and the item revision specialist. After an item was initially drafted by the author according to predefined requirements, it was independently evaluated by three reviewers. These reviewers worked separately without communication and provided individual feedback. All review comments were then compiled and delivered to the item revision specialist, who modified the items based on the reviewers' suggestions and conducted an additional review before finalization. This workflow was intentionally designed to have a dedicated revision specialist, rather than the original author, perform modifications to mitigate potential errors arising from the author's cognitive biases or misconceptions. In addition, independent input from three reviewers allowed the revision specialist to cross-validate the feedback, thus reducing the risk of erroneous or overlooked revisions.

After being subjected to four rounds of review, the items were finalized and incorporated into an internal item bank. From this repository, we selected 363 items characterized by high discriminatory power and strong reasoning demands. These items are typically located within authentic biological research contexts and require respondents to integrate biological knowledge with logical reasoning to formulate correct answers. During the selection process, we did not refer to any AI-generated responses to these items. Upon completion of the screening, human experts were again invited to examine the selected items, which were ultimately designated as the test set for this study.

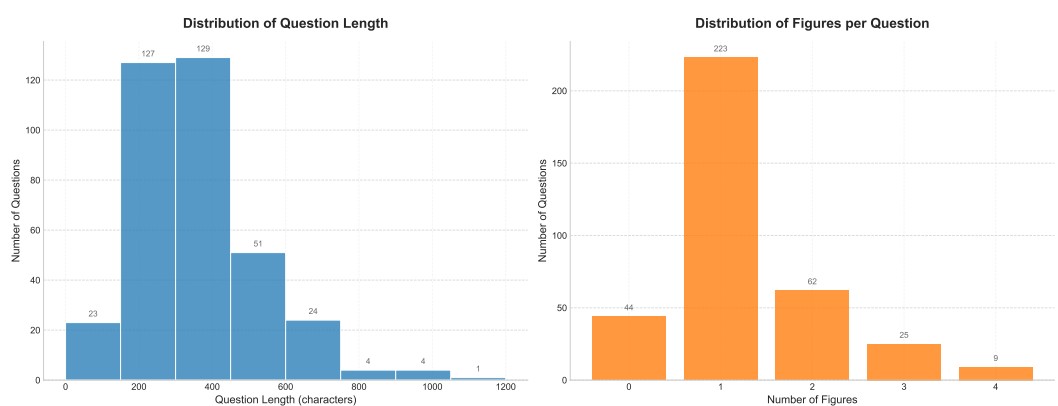

Figure 2: Distributions of question lengths and figure counts in OlymBio-Bench.

We analyzed the distribution of item lengths and the figure counts in the test set, with the results presented in Figure 2. Additionally, we quantified the proportion of items from various biological subdisciplines included in the test set, as shown in Figure 3.

It can be observed that coverage across disciplines is relatively even, enabling a well-balanced evaluation of the model's capabilities without strong bias toward any particular subfield. This is a key advantage of OlymBio-Bench over HLE and other benchmarks: it spans a more comprehensive range of biological subjects. Moreover, many terms are inherently interdisciplinary—for example, items in Genetics and Evolutionary Biology require models to combine genetic knowledge with an evolutionary background—making the benchmark better suited to assessing a model's ability to integrate knowledge for reasoning.

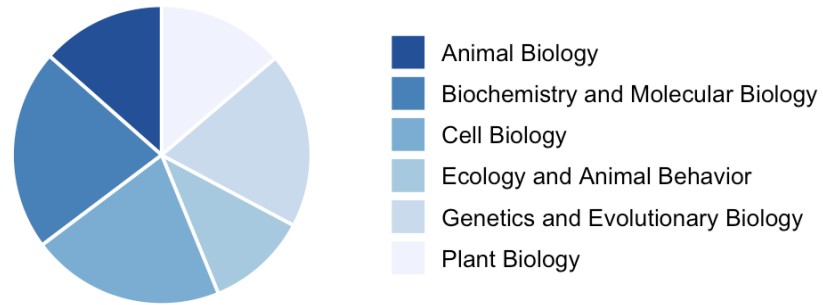

Figure 3: Distribution of Questions by Biological Disciplines.

## 3.2 BASELINE MODELS & PROMPT

Given that the majority of questions in the OlymBio-Bench benchmark are image-based, our study primarily utilized Vision-Language Models (VLMs). The evaluated models include the GPT-4.1, GPT-o4-mini, GPT-4, and GPT-5 series; the Gemini-2.5 series; the open-source Qwen-2.5-VL series; and Grok-4.

We conducted the evaluation using a zero-shot prompting methodology. The prompt is shown below:

> Evaluate each of the following statements, determining if it is True or False, please provide step-by-step solutions in your output.
>
> Your final answer must be a single string consisting only of the letters 'T' (for True) and 'F' (for False), with no spaces or other characters. The order of the letters in your response must directly correspond to the order of the statements. Put your final answer in **** and do not use * in any other ways.
>
> For example, if there are four statements and you find the first to be True, the second False, the third False, and the fourth True, your answer must be exactly:
>
> ```
> ****TFFT****
> ```
>
> {Question}

We employ regular expressions to parse the outputs and extract the answers, which are then compared against the ground truth to calculate the accuracy. Specifically, we adopt an exact match criterion, where a prediction is counted as correct only if it matches the ground truth exactly.

## 4 RESULTS

### 4.1 SOTA MODELS STILL FAIL TO ACHIEVE A PASSING SCORE ON OLYMBIO-BENCH

We evaluated a set of state-of-the-art LLMs—including the Gemini 2.5 family, OpenAI's recent series (e.g., GPT-5 and its efficient variants, o4-mini), Qwen 2.5, and xAI's Grok—on OlymBio-Bench. As shown in Table 1, no model reaches a passing score of $60\%$; the best result is achieved by `gemini-2.5-pro` at $51.79\%$, underscoring persistent limitations of current systems on challenging biology questions.

Among the strongest performers are `gemini-2.5-pro` and `gpt-5`. Both are most up-tpo-date reasoning-oriented models and achieve roughly $\sim 50\%$ on multimodal tasks and $\sim 60\%$ on text-only tasks, with `gemini-2.5-pro` slightly higher overall. Other reasoning models—`gpt-5-mini`, `o4-mini-high`, `o4-mini`, and `grok-4`—follow, scoring $42\%$–$46\%$. In general, instruction-tuned (non-reasoning) models rank below reasoning counterparts, indicating that *OlymBio-Bench* effectively differentiates models by their reasoning capacity.

Across most models, accuracy on text-only items exceeds that on multimodal items. This modality gap suggests that incorporating visual evidence—central to biological practice—remains difficult. Notably, inputs such as microscopy images, electrophoresis gel images, and statistical plots are commonplace and essential in biology, so our benchmark emphasizes multimodal evaluation to reflect real-world analytic demands. The results highlight a substantial performance gap for expert-level biological reasoning.

Finally, within model families (e.g., Gemini 2.5, GPT-5), we observe a positive correlation between parameter scale and accuracy on OlymBio-Bench. This trend supports that the benchmark rewards genuine reasoning rather than guesswork. In sum, OlymBio-Bench is a multimodal assessment with high difficulty that exposes current limitations and provides resolution to compare reasoning capabilities across modern LLMs.

## 5 DISCUSSION

### 5.1 DEFICIENCIES IN REASONING AND MULTI-MODAL UNDERSTANDING AS MAJOR CAUSES OF ERRORS

OlymBio-Bench is a scientific benchmark that involves multi-modal information. To provide a detailed analysis of its difficulty, we examined the accuracy of each model on questions with varying lengths and figure counts, as depicted in Figure 4. We found that as the question length and figure count increase, nearly all models exhibit a consistent downward trend in accuracy, reflecting the challenges posed to LLMs by the complexity of reasoning and image analysis. An interesting

Table 1: Model Performance (Sorted by Accuracy, Total %)

| Model | Total (363) | | Multi-modal (319) | | Text-only (44) | |
|---|---|---|---|---|---|---|
| | Count | % | Count | % | Count | % |
| gemini-2.5-pro (Comanici et al., 2025) | 188 | 51.79% | 161 | 50.47% | 27 | 61.36% |
| gpt-5(OpenAI, 2025b) | 185 | 50.96% | 159 | 49.84% | 26 | 59.09% |
| gpt-5-mini(OpenAI, 2025b) | 168 | 46.28% | 142 | 44.51% | 26 | 59.09% |
| o4-mini-high(OpenAI, 2025c) | 167 | 46.01% | 141 | 44.20% | 26 | 59.09% |
| o4-mini(OpenAI, 2025c) | 162 | 44.63% | 140 | 43.89% | 22 | 50.00% |
| grok-4(xAI, 2025) | 156 | 42.98% | 131 | 41.07% | 25 | 56.82% |
| gpt-4.1(OpenAI, 2025a) | 155 | 42.70% | 130 | 40.75% | 25 | 56.82% |
| chatgpt-4o-latest(OpenAI, 2024) | 147 | 40.50% | 125 | 39.18% | 22 | 50.00% |
| gpt-4.1-mini(OpenAI, 2025a) | 137 | 37.74% | 114 | 35.74% | 23 | 52.27% |
| gemini-2.5-flash(Comanici et al., 2025) | 134 | 36.91% | 114 | 35.74% | 20 | 45.45% |
| gpt-5-nano(OpenAI, 2025b) | 113 | 31.13% | 93 | 29.15% | 20 | 45.45% |
| gemini-2.5-flash-lite(Comanici et al., 2025) | 107 | 29.48% | 87 | 27.27% | 20 | 45.45% |
| qwen_qwen2.5-vl-72b-instruct(Bai et al., 2025) | 99 | 27.27% | 83 | 26.02% | 16 | 36.36% |
| qwen_qwen2.5-vl-32b-instruct(Bai et al., 2025) | 95 | 26.17% | 75 | 23.51% | 20 | 45.45% |
| gpt-4.1-nano(OpenAI, 2025a) | 59 | 16.25% | 46 | 14.42% | 13 | 29.55% |

observation regarding the effect of question length on accuracy is that top-performing models (e.g., gemini-2.5-pro, gpt-5) typically exhibit a negative correlation with question length. In contrast, lower-performing models (e.g., qwen_qwen2.5-vl-32b-instruct, gpt-4.1-nano) are less sensitive or even show a slight positive correlation. Regarding the introduction of figures, we observed that all models experience a drop in accuracy when one figure is introduced (n=0 vs. n=1). However, as the number of figures increases further, the subsequent decline in accuracy becomes more gradual. Futhermore, we point out that the drop in accuracy is more pronounced with an increased number of figures than with increased question length. This indicates that analyzing image content is more difficult for the models and may be a critical area for future breakthroughs in the development of LLMs.

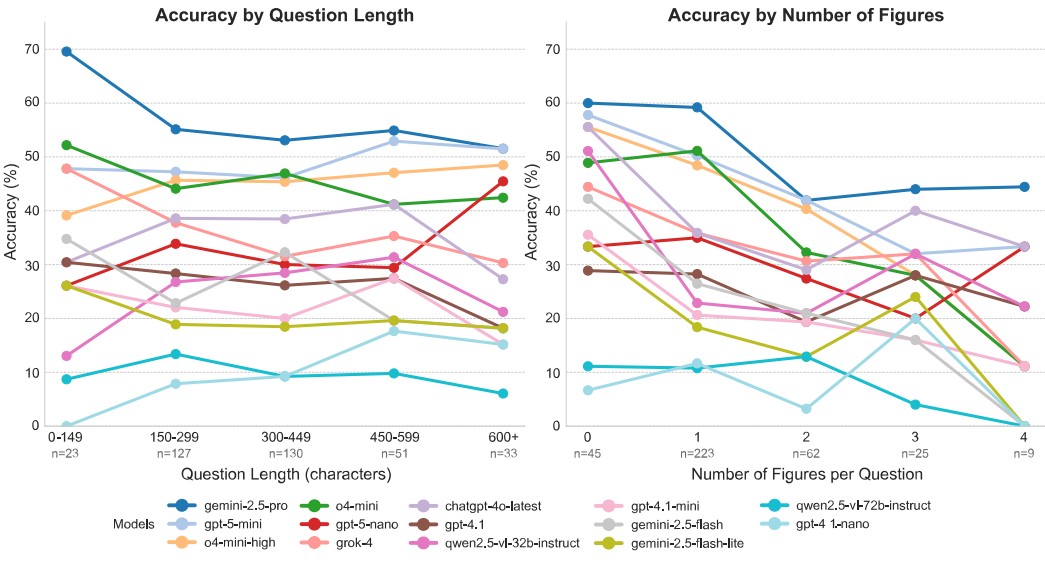

Figure 4: Correlation of LLM accuracy on OlymBio-Bench with question length and figure count.

We selected the best-performing models, gemini-2.5-pro and gpt-5, and the worst-performing model, gpt-4.1-nano, to conduct an error analysis on 38 multi-modal questions that were answered incorrectly by all 15 models (Figure 5). The results show that the vast majority ( 80%) of errors were caused by incorrect comprehension of or information extraction from the images. Furthermore, gpt-4.1-nano produced a few invalid outputs due to formatting errors, indicating its weaker instruction-following capabilities. In summary, we posit that when solving complex, real-world

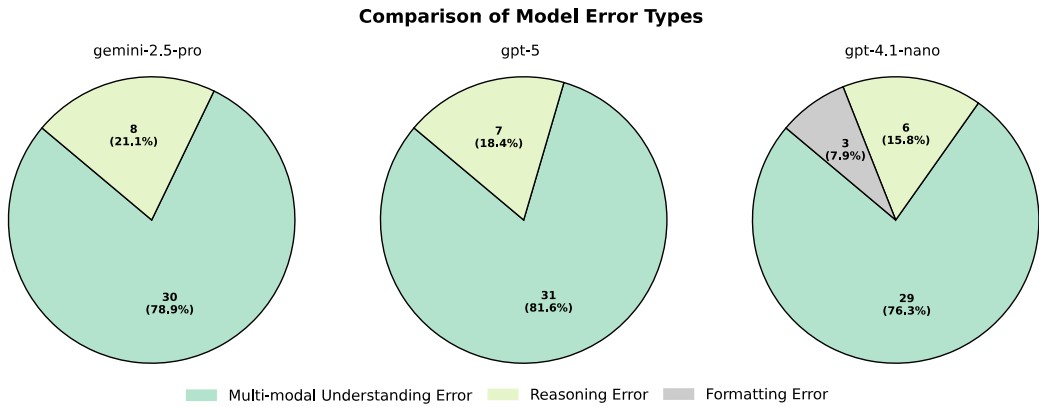

Figure 5: Error Types across various models on multi-modal questions.

biological problems, the current ability of LLMs to understand multi-modal information, such as images, is insufficient and consequently limits their performance in the biological domain.

## 6 CONCLUSION

All in all, we have constructed a biological benchmark that features high discriminatory power, high difficulty, and is close to the frontiers of life science research. Previously, due to the relatively low difficulty of the benchmark related to biological reasoning questions, many large language models achieved high scores, which made it difficult for us to effectively evaluate the comprehension abilities of these large language models when dealing with biological problems. This newly constructed benchmark has solved the problems of insufficient difficulty and lack of reasoning in the existing biological benchmark.

Among the models we tested, even the best-performing model only answered approximately half of the questions correctly. The low accuracy reveals that there is still substantial room for improvement in the problem-solving abilities of current large language models in the context of biological scientific research, providing a good reference framework for strengthening the biological comprehension and problem-solving abilities of large language models.

Meanwhile, we also analyzed the causes of errors made by large language models and found that most of the errors can be attributed to the deficiencies in their multi-modal capabilities. This implies that current large language models still have flaws in their comprehension abilities regarding the cutting-edge scientific research charts in biology, which also provides some clues for further enhancing the biological problem-solving abilities of large language models.

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

# A APPENDIX

## A.1 EXAMPLE QUESTIONS FROM THE DATASTE

Here we show one example from each subdiscpline in our dataset.

---

**Animal Biology**

P57 is a natural bioactive compound extracted from Hoodia gordonii that can markedly suppress appetite in animals. Researchers administered intraperitoneal injections of P57 at varying concentrations in mice (Vehicle denotes solvent-only injection) and recorded food intake over 72 hours, as shown in the figure. They also observed that, over a short time frame, mice injected with P57 exhibited a decrease in body temperature compared with mice receiving Vehicle only(Wang et al., 2023). Judge the correctness of the following statements:

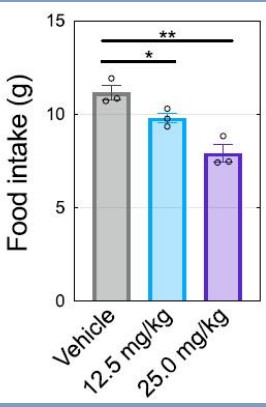

A) In the short period after injection, the body temperature of mice receiving 20 mg/kg P57 may be higher than that of mice receiving 15 mg/kg P57.

B) P57 may lead to a reduction in the overall metabolic rate of mouse cells.

C) Injecting an appropriate concentration of P57 may enable the establishment of a mouse model that mimics hibernation.

D) The microscopic mechanism by which P57 suppresses appetite requires follow-up biochemical analyses for elucidation.

---

**Biochemistry and Molecular Biology**

Transient DNA:RNA hybridizations are an important step in gene expression but can also lead to DNA damage. scRad27 is known to be a yeast nuclease that participates in joining Okazaki fragments, and RAD27-AID is a factor that can specifically degrade scRad27. S9.6 is a monoclonal antibody that specifically binds DNA:RNA hybrids, while anti-dsDNA is a monoclonal antibody that specifically binds double-stranded DNA. In the presence or absence of Auxin, researchers measured the cellular level of scRad27 (panel b), used S9.6 and anti-dsDNA to assess the amounts of DNA: RNA hybrids and double-stranded DNA (panel c) were further analyzed, as well as the relative level of DNA:RNA hybrids (panel d)(Mangione et al., 2025). Judge whether the following statements are true or false (T/F):

---

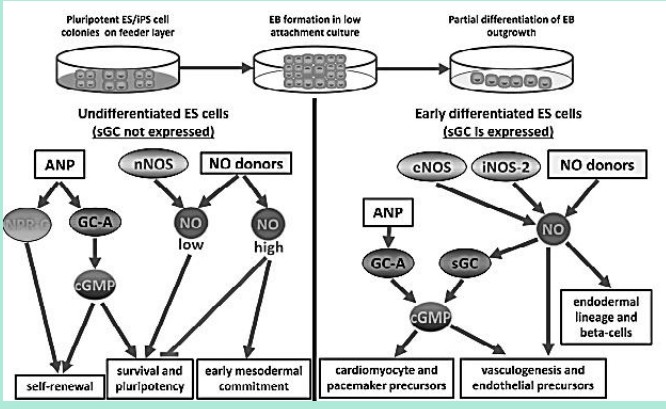

A) RAD27-AID can effectively degrade scRad27 under any condition.

B) scRad27 can hydrolyze the RNA primer segments used to join Okazaki fragments, thereby allowing the DNA of adjacent Okazaki fragments to be correctly ligated.

C) If the gene encoding scRad27 is inactivated, the cellular level of DNA:RNA hybrids in yeast is likely to decrease.

D) Under normal conditions, scRad27 may reduce the likelihood of DNA damage.

## Cell Biology

Both cGMP and NO are key signaling molecules in cellular signal transduction pathways. To investigate their roles in inducing cellular differentiation, scientists treated embryonic stem cells (ES) and induced pluripotent stem cells (iPS), as well as early differentiating cells obtained after these stem cells developed into embryoid bodies (EBs), with atrial natriuretic peptide (ANP) and several nitric oxide synthase (NOS) isoforms. They then measured intracellular activities and analyzed the signaling pathways; the results are shown in the figure. Annotations: NPR-C: a type of ANP receptor, a member of the G protein–coupled receptor superfamily; GC-A: guanylyl cyclase A; sGC: soluble guanylyl cyclase; nNOS: neuronal NOS; eNOS: endothelial NOS; iNOS-2: inducible NOS-2. self-renewal; survival and pluripotency; early mesodermal commitment; cardiomyocyte and pacemaker precursors; vasculogenesis and endothelial precursors; endodermal lineage and beta-cells(Kots & Bian, 2024). Judge the correctness (true/false) of the following statements:

A) Because ANP is synthesized in the atria, ANP has no effect on the differentiation of cardiovascular-related cell types in early embryos.

B) As cells differentiate, sGC-related genes begin to be expressed, and the predominant NOS isoform also changes.

C) NO may play an important role in cell differentiation, and its function may vary across different stages of embryonic cell differentiation.

D) Based on the above results, it can be inferred that sGC may be an important factor linking the transduction of ANP and NO signals.

### Ecology and Animal Behavior

Ecologists recorded changes in the biomass density of Centrostephanus rodgersii (a sea urchin that primarily grazes on macroalgae) across several areas of Tasmania's east coast from 2009 to 2023. The results are shown in the figure: the dashed line denotes the ecologically expected optimal biomass density for sea urchins; the dotted polyline before 2023 represents the observed sea urchin biomass density; and the lines after 2023 indicate projected biomass density under different fishing pressures. H denotes the harvest intensity on sea urchins, where H = 0 indicates no fishing and H = 1.00 indicates harvesting all catchable urchins. Note: In all three panels, among the radial projection lines extending beyond 2023, trajectories lower on the plot correspond to H values approaching 1.00, whereas higher trajectories correspond to H values approaching 0(Cresswell et al., 2025). Judge the correctness (true/false) of the following statements:

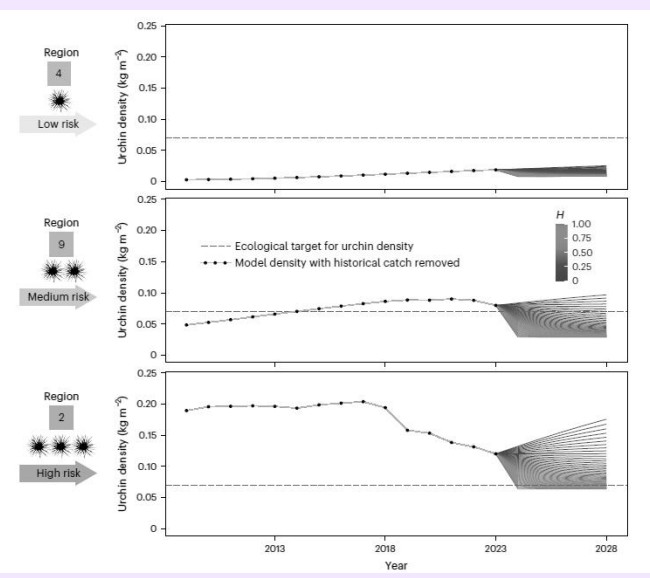

A) *C. rodgersii* is a naturally invasive species on Tasmania's east coast.

B) An overabundance of sea urchins may degrade the local kelp-forest ecosystem.

C) Heavy harvesting of sea urchins will inevitably disrupt the balance of the local ecosystem.

D) As sea urchin population density increases, the expected differences in population outcomes under varying harvest intensities also increase.

### Genetics and Evolutionary Biology

Researchers inserted a genomic segment from a sponge into the zebrafish genome. This segment contains two genes, *lsl* and *Scaper*. An enhancer is located within one of these genes. In zebrafish, the expression patterns driven by the sponge sequence and by the endogenous zebrafish genes are shown in the figure. Note: zf = zebrafish; UIC = uninjected control; sponge = sponge-derived sequence(Wong et al., 2020). Judge the correctness (true/false) of the following statements:

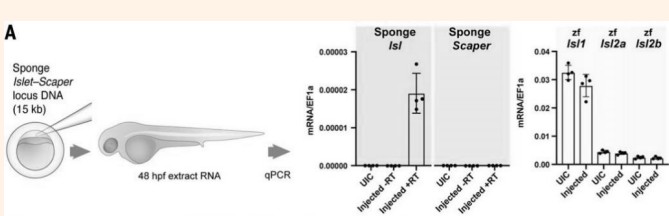

A) The enhancer for the sponge *lsl* gene resides within the lsl gene itself.

B) It can be inferred that the sponge gene segment contains a transcription factor that increases the expression level of the zebrafish *lsl1* gene.

C) This experiment provides evidence for deep conservation of gene regulatory mechanisms across the animal kingdom during metazoan evolution.

D) As sea urchin population density increases, the expected differences in population outcomes under varying harvest intensities also increase.

### Plant Biology

ABA DEFICIENT 2 (*ABA2*) and BETA-GLUCOSIDASE 1 (*BG1*) are key genes in abscisic acid (ABA) biosynthesis. Compared with the wild type, the *aba2* and *bg1* mutants show markedly reduced anthocyanin accumulation under low-phosphate (LP) conditions. LP-induced anthocyanin accumulation is likewise reduced in ABA-signaling mutants. Overexpression of the transcription factor ABI5 rescues the LP-induced anthocyanin phenotype in these ABA-related mutants, and ABI5 expression is significantly decreased in ABA-signaling mutants(Song et al., 2024). Judge whether the following statements are true or false.

A) Stomata are abnormally open in ABA-biosynthesis mutants.

B) Overexpression of *ABA2* reduces SnRK2 activity, whereas *ABA2* deficiency decreases PP2C activity.

C) ABI5 mediates ABA-promoted anthocyanin accumulation under LP.

D) Introducing the structural gene CHS together with its cognate regulatory module in the anthocyanin pathway restores the *bg1* anthocyanin-accumulation phenotype to wild-type levels.

