# OpenReview forum: "OlymBio-Bench: A Multimodal Challenge Towards Expert-Level Biological Reasoning"
_ICLR.cc/2026/Conference — Submitted to ICLR 2026_

### Official Review · Reviewer_iFPm · 2025-10-31

**Soundness:** 2
**Presentation:** 2
**Contribution:** 1
**Rating:** 0
**Confidence:** 5

**Summary:**

This paper introduces  OlymBio-Bench, a multimodal benchmark for biology sourced from 220 research papers in life sciences and curated by a team of 60 authors and reviewers. The benchmark is used to evaluate a range of state-of-the-art models. Results reveal that sota models fail to achieve a passing score, highlighting critical deficiencies in their ability to perform complex, multimodal scientific reasoning.

**Strengths:**

The work addresses a relevant and important problem.

The authors evaluate several state-of-the-art language and multimodal models on the benchmark

**Weaknesses:**

**1.- Disregards previous literature**:

This work disregards prior relevant benchmarks. Specifically, it does not sufficiently acknowledge or position itself relative to recent unimodal and multimodal biology benchmarks such as MicroBench, MicroVQA, LAB-Bench, BioASQ, just to name a few. Without a clear comparative analysis or discussion of how the proposed benchmark differs from and improves upon these prior efforts, the novelty and incremental contribution of this work remain unclear.

**2. This benchmark appears to focus primarily on chart, diagram, and plot interpretation rather than biological visual understanding.** None of the example questions in the appendix include images of biological specimens at either microscopic or macroscopic resolution. In cell biology, researchers routinely rely on diverse microscopy modalities (e.g., fluorescence microscopy, light microscopy, electron microscopy) to study biological systems. These imaging approaches are foundational to modern biology, yet this benchmark does not include such modalities. As a result, it is difficult to characterize this as a comprehensive “biology” benchmark when one of the core pillars of biological research (microscopy) is entirely absent.

**suggestion**: Expand this benchmark to include microscopy.

**3. Images are not required to answer the questions:** The example questions suggest that images are largely unnecessary for solving the tasks. For instance, in the first example (Animal Biology), the textual description alone appears sufficient to answer the question, indicating that visual inputs are superficial rather than essential. This raises questions about whether the benchmark meaningfully evaluates multimodal biological reasoning.

**Suggestion:** Quantify the proportion of “multimodal” questions that can be answered correctly without access to the image, and report this statistic. This would help demonstrate that the benchmark genuinely requires visual understanding rather than relying primarily on text. I also recommend reviewing the Cambria benchmark and considering mechanisms used there to ensure that tasks are truly vision-centric; similar design principles could strengthen this work.

**4. Missing essential dataset descriptors:** The paper omits key dataset details. For instance, the total number of questions in the benchmark is not clearly stated anywhere in the manuscript. Based on Table 1, one might infer that the dataset contains approximately 1,000 questions, but this should be explicitly reported. Clear dataset statistics are critical for evaluating the scale and significance of the benchmark.

**Suggestion**: Please provide more datset stats.

**Questions:**

Suggestions are shared above ^

---

### Official Review · Reviewer_aBYe · 2025-10-31

**Soundness:** 2
**Presentation:** 2
**Contribution:** 2
**Rating:** 2
**Confidence:** 3

**Summary:**

This paper introduces OlymBio-Bench, an expert-level multimodal benchmark for biology. The benchmark contains 363 problems, most of them multimodal. It spans multiple subfields in biology. Questions are sourced from practice sets originally developed for participants in International Biology Olympiad. Evaluation shows that state-of-the-art models do not perform well on the benchmark.

**Strengths:**

+ The fact that state-of-the-art models underperform on OlymBio-Bench shows that it is of sufficient difficulty and points to weaknesses in current models.
+ Its focus on \textbf{multimodal} biology problems is a strength, since they are difficult to source.

**Weaknesses:**

- The abstract claims questions are "sourced from over 220 frontier research papers in life sciences" but Section 3.1 says "All questions were sourced from practice sets originally developed for participants in International Biology Olympiad". This seems conflicting. Where do the questions come from exactly?
- 363 questions is on the lower side for a benchmark.
- As a benchmark paper, data quality is the most important factor for consideration. But this paper only includes a few examples. It would be better if there was a link to an anonymous site hosting the data for review.
- The paper currently stands at only 7 pages. A lot more details on curation, dataset examples & analysis, and error analysis can be added. I also find the error analysis to be a little superficial.

**Questions:**

- Rules permitting, would it be possible to provide a link to your an anonymized repository of the questions?
- Could the authors provide additional details on dataset source and curation?

---

### Official Review · Reviewer_NMuL · 2025-11-01

**Soundness:** 3
**Presentation:** 2
**Contribution:** 1
**Rating:** 2
**Confidence:** 3

**Summary:**

The paper presents **OlymBio‑Bench**, an expert‑level multimodal biology benchmark constructed from IBO‑style problems, emphasizing integrated reasoning over text and figures. The dataset contains **363 items** spanning five subdisciplines, with most items including at least one figure (Sec. 3.1, p.3; Fig. 2–3, p.4). Models (Gemini‑2.5, GPT‑5 family, Grok‑4, Qwen‑2.5‑VL) are evaluated zero‑shot with an exact‑match scoring pipeline based on a fixed final‑answer string (Sec. 3.2, p.5). **No model passes 60%;** the best is gemini‑2.5‑pro at 51.79% overall, with text‑only items generally easier than multimodal ones (Table 1, p.6; Sec. 4.1, p.5–6).

**Strengths:**

1. **Expert‑level multimodal focus**
- Majority of items are image‑dependent, aligning with authentic biological analysis (Fig. 2 right, p.4), which raises ecological validity for VLM evaluation.
- Breadth across five subdisciplines supports cross‑topic reasoning (Fig. 3, p.4), increasing coverage beyond narrow tasks.
- Examples in the appendix tie questions to research‑style scenarios, illustrating realism (A.1, pp.10–13).
2. **Documented curation pipeline**
- Multi‑stage process with three independent reviews plus a revision specialist (Fig. 1; Sec. 3.1, p.3) supports quality control and reduces single‑author bias.
- Final selection of 363 items targets “high discriminatory power and strong reasoning demands” (Sec. 3.1, p.3), aligning with aims.
- Additional human expert inspection after sampling provides an extra QA layer (Sec. 3.1, p.3).

**Weaknesses:**

1. **Similarity to existing biology benchmarks is under‑analyzed**
- The paper itself cites LAB‑Bench (multimodal figure/table interpretation and protocol tasks), VCT (multimodal virology), and Bix‑Bench (computational‑biology agents) but does not systematically differentiate OlymBio‑Bench from these resources beyond general claims (Sec. 2.2, pp.2–3). This leaves novelty vs. prior art ambiguous.
- No side‑by‑side comparison (tasks, item formats, difficulty, modality mix) or cross‑benchmark evaluation/transfer study is provided—e.g., overlap with LAB‑Bench FigQA/TableQA is asserted qualitatively but not quantified (Sec. 2.2, pp.2–3). No direct evidence found in the manuscript.
- The claim that OlymBio covers “a more comprehensive range of biological subjects” than HLE is not accompanied by a definition of HLE or a tabulated coverage comparison (p.4). This reduces clarity of the distinct contribution.
2. **Release and confidentiality are in tension**
- The Abstract/Intro says “We release OlymBio‑Bench,” while Sec. 3.1 states questions come from IBO practice sets under confidentiality and are “not publicly available,” creating ambiguity about what is actually released (Abstract, p.2; Sec. 3.1, p.3).
3. **Under‑specified evaluation setup**
- Sampling parameters (temperature, top‑p, seed, max tokens) for each API are not reported (Sec. 3.2, p.5). No direct evidence found in the manuscript. Image preprocessing (resolution, cropping) is not described, though core to multimodal performance (Sec. 3.2, p.5). No direct evidence found in the manuscript. The paper’s own error analysis notes formatting errors in weaker models, showing the brittleness of regex‑only parsing (Fig. 5, p.7; Sec. 3.2, p.5).

**Questions:**

**Novelty vs. prior art:** Please provide a side‑by‑side comparison with LAB‑Bench, VCT, and Bix‑Bench (task types, item format, modality mix, difficulty) and define HLE to substantiate distinct contributions (Sec. 2.2, pp.2–3; p.4).

---

### Official Review · Reviewer_AMLL · 2025-11-06

**Soundness:** 2
**Presentation:** 3
**Contribution:** 2
**Rating:** 4
**Confidence:** 2

**Summary:**

The paper introduces OlymBio-Bench, a multimodal benchmark designed to evaluate expert-level biological reasoning. The dataset consists of 363 curated problems derived from International Biology Olympiad (IBO) materials and recent research topics. Each question is reviewed through a multi-stage pipeline involving item authors, multiple reviewers, and a revision specialist. Experiments show that even the strongest models fail to reach human-level accuracy with performance dropping significantly on multimodal and longer questions.

**Strengths:**

1.The dataset’s construction pipeline comprises authoring, triple-reviewing, and expert revision, ensuring reliability and conceptual correctness, reducing annotation bias and improving scientific validity.
2.The benchmark effectively exposes current LMMs’ limitations in visual-biological reasoning. The observed performance degradation with increased question length or image complexity provides actionable insights into model weaknesses.

**Weaknesses:**

1.Insufficient transparency in data collection and annotation. Although the paper mentions multiple roles (authors, reviewers, revision specialists), it lacks key details: the total number of contributors in each role, their expertise levels, and how reviewer feedback was structured and incorporated. Annotation criteria and conflict-resolution protocols are not clearly defined.
2.Limited dataset statistics and diversity analysis. Beyond distributions of question length and number of images, the paper does not quantify diversity across question types or the variety of image modalities.
3.Unclear answer format. It remains unclear whether the evaluation is limited to multiple-choice questions or also includes short-answer and open-ended formats.
4.Lack of comparison with existing benchmarks. While the paper discusses general benchmarks (e.g., GPQA), it does not provide empirical comparisons or difficulty calibration against them.
5.Limited evaluation depth. Current analysis focuses mainly on modality (image vs. text). The paper should include breakdowns by biological subfields and cross-category performance, as well as tests with research-oriented agents (e.g., OpenDeepResearch [1]) to better reflect real scientific reasoning capabilities.

**Questions:**

please refer to weakness

---

### Meta-Review · Area_Chair_rUXq · 2025-12-31

**Summary:**

This paper introduces OlymBio-Bench, an expert-level multimodal benchmark for biology. The benchmark contains 363 problems, most of them multimodal. It spans multiple subfields in biology. Questions are sourced from practice sets originally developed for participants in the International Biology Olympiad. Evaluation shows that state-of-the-art models do not perform well on the benchmark.

**Reviewer Concerns:**

1. Similarity to existing biology benchmarks is under‑analyzed
2. Under‑specified evaluation setup, the empirical study is not solid.
3. Missing essential dataset descriptors and open source plan are not clear.

**Reviewer Scores:**

The authors have not submitted rebuttals. These concerns, especially the major ones, still hold. AC made the recommendation of rejection.

---

### Decision · Program_Chairs · 2026-01-26

Reject